# Citrus Pruning in the Mediterranean Climate: A Review

**DOI:** 10.3390/plants12193360

**Published:** 2023-09-22

**Authors:** Pedro Matias, Isabel Barrote, Gonçalo Azinheira, Alberto Continella, Amílcar Duarte

**Affiliations:** 1Faculdade de Ciências e Tecnologia, Campus de Gambelas, Universidade do Algarve, 8005-139 Faro, Portugal; pmmatias@ualg.pt (P.M.); ibarrote@ualg.pt (I.B.); 2MED—Mediterranean Institute for Agriculture, Environment and Development and CHANGE—Global Change and Sustainability Institute, Faculty of Sciences and Technology, Campus de Gambelas, Universidade do Algarve, 8005-139 Faro, Portugal; 3Centre of Marine Sciences (CCMAR), Campus de Gambelas, Universidade do Algarve, 8005-139 Faro, Portugal; 4Instituto Superior de Engenharia, Campus da Penha, Universidade do Algarve, 8005-139 Faro, Portugal; gjazinheira@ualg.pt; 5Department of Agriculture, Food and Environment, University of Catania, Via S. Sofia 100, 95131 Catania, Italy; acontine@unict.it

**Keywords:** canopy management, training system, formative pruning, maintenance pruning, mechanical pruning, alternate bearing, mandarin

## Abstract

Pruning is a common practice in citrus for various reasons. These include controlling and shaping the canopy; improving phytosanitary health, productivity, and fruit quality; and facilitating operations such as harvesting and phytosanitary treatments. Because pruning is an expensive operation, its need is sometimes questioned. However, it has been proven to be particularly important in Mediterranean citriculture, which is oriented towards producing fruits for a high-quality demanding fresh market. Herein, we summarize and explain the pruning techniques used in Mediterranean citriculture and refer to the main purposes of each pruning type, considering citrus morphology and physiology.

## 1. Introduction

Pruning is a common cultural practice in citrus and one of the most expensive orchard maintenance operations. Even so, technical information on citrus pruning is relatively scarce, especially in peer-reviewed scientific journals.

In the Scopus database, there are only 70 articles on citrus pruning [Search by: TITLE-ABS-KEY “citrus” OR “mandarin” OR “poncirus” AND TITLE “pruning”]. In this database, a yearly average of only four papers were published in the last ten years; none of these was a review. Therefore, a review of the cutting methods and the needs of the main citrus species and cultivars and a critical analysis of pruning are needed to put together and summarize the current knowledge on and technical aspects of citrus pruning.

Citrus fruits are cultivated worldwide, but the technologies that can be used depend on the edaphoclimatic conditions and production objectives. The Mediterranean basin is a subtropical area where citrus have adapted and have significant economic and cultural importance. Moreover, citrus are now an essential part of the Mediterranean landscape and diet [1,2].

This review will address general aspects of citrus pruning. It will focus, in more detail, on the practices used in Mediterranean countries, according to the authors’ experiences after several years of work on pruning trials.

### 1.1. Pruning Origin

Although pruning is an ancestral practice, it originated many years after agriculture was consolidated as a major activity of humankind. Reports from the time of Ancient Greece indicate that the first “pruning” was performed by livestock (donkeys, sheep, goats) and by meteorological events (wind and hailstorms). Farmers began to observe that these “pruned” trees behaved differently from unpruned trees. As a result, farmers tried to reproduce this livestock-based and meteorological “pruning” artificially. From then on, knowledge continued to evolve, and pruning is now considered an essential practice for many crops [3,4].

### 1.2. Pruning Definition

Pruning consists of the removal/suppression of plant parts such as branches, roots, and buds. Some authors limit the definition of pruning to the removal of vegetative or fruit-bearing organs, provided that such removal affects the physiological behavior of the plant in question [5]. Pruning allows the development control of the tree and delays its growth.

## 2. General Aspects of Citrus Morphology and Physiology

Pruning allows tree growth management and development using cuts and affects the plant’s physiological behavior [6]. Therefore, understanding the general aspects of morphology and physiology is essential for designing appropriate pruning actions.

### 2.1. Citrus Phenological Cycle

The plant’s phenological cycle must always be considered when planning to prune it; thus, knowing the stages of the citrus plant’s phenological cycle, such as shoot growth, flowering, and fruit development, is fundamental (Figure 1).

#### 2.1.1. Shoot Formation and Flowering

In citrus, flowering and vegetative growth depend on the formation of shoots, which arise from apical and axillary buds. Shoots can be of five different types, as illustrated in Figure 2: (a) Multi-flower mixed shoots (MFM), with more than one flower and at least one leaf; (b) single-flower mixed shoots (SFM), with only one terminal flower and at least one leaf; (c) multi-flower generative shoots (MFG), with more than one flower and no leaves; (d) single-flower generative shoots (SFG), with one single flower and no leaves; and (e) vegetative shoots (V), without flowers and only with at least one leaf. Types (a) and (b) can be grouped under the designation of mixed shoots, and types (c) and (d) are both generically called generative shoots.

In subtropical climates, there are usually three periods of shoot growth per year: spring, summer, and autumn (Figure 1). Flowering is promoted by the low winter temperatures; in most cultivars, the plant blooms only in spring. In summer and autumn, only vegetative growth occurs. Nonetheless, there are some everblooming cultivars wherein flowering occurs in all shoot growth flushes, such as some lemon and lime cultivars [7,8]. Pruning may lead to earlier shoot formation [9,10,11]. The application of growth regulators can also affect shoot growth flushes [12].

In subtropical climates, flowering and other related aspects can be used to group cultivars according to their bearing habits: (i) single-annual-bearing cultivars (SB), with regular production only once a year, every year; (ii) multiple-annual-bearing cultivars (MB) that produce more than once a year, every year; and (iii) alternate-bearing cultivars (AB) with abundant yields one year and scarce or no yields the next.

The shoots formed in spring develop from the axillary buds of the previous year’s branches that were formed in summer and autumn (Figure 3) [7,13]. At sprouting, one or more shoots can emerge from each node. The proportion of nodes where the buds remain dormant is higher in the older branches [7,14].

#### 2.1.2. Fruit Development

Fruit development occurs after flowering and fruit set. Fruit development has three main stages (Figure 1) [15]: (i) Stage I—intense cell division takes place, and fruit growth is slow; (ii) Stage II—cell expansion, and fruit growth is fast; and (iii) Stage III—lower fruit growth rate, and fruit maturation occurs.

The fruit development period depends on the species and cultivar (Figure 1), which can be of three types: (i) early maturing cultivars, for which the fruit development period takes seven months or less (harvesting is carried out more than four months before the next year’s flowering); (ii) mid-season cultivars, for which the fruit development period is between 8 and 12 months (harvesting for year one is carried out before the flowering in year two, or slightly later when delayed); and (iii) late maturing cultivars, for which the fruit development period is longer than 12 months (harvesting for year one is carried out one to three months after the flowering in year two).

### 2.2. Shooting Habit

Shooting habit refers to the way trees form new shoots. According to this, citrus cultivars can be classified on the basis of three shooting habits (Figure 4): (a) short multiple shoots (SMSs), (b) intermediate shoots (ISs), and (c) long solitary shoots (LSSs).

SMS cultivars form short and usually multiple shoots (more than one shoot per node) (Figure 4a). Mediterranean mandarin (‘Setubalense’, ‘Avana’, and other cultivars) and the clementine group are examples of SMS-shooting-habit trees. These cultivars tend to form dense canopies with very dense foliage.

LSS shooting cultivars form long, usually solitary shoots (only one shoot per node) (Figure 4c). Lemon cultivars and the satsuma group are examples of LSS cultivars. These cultivars tend to form sparse canopies with scattered foliage.

IS cultivars form shoots that are not as long as in LSSs but are longer than those of SMS cultivars; sometimes, they form more than one shoot per node (Figure 4b). Most orange tree cultivars are IS cultivars. These cultivars have less dense canopies than SMS cultivars and are more compact than LSS cultivars.

There are also cultivars exhibiting characteristics that lie somewhere between these main types of shooting habits, and not all the shoots on a tree are characteristic of that shooting type of cultivar.

### 2.3. Tree Growth—Formation, Distribution, and Accumulation of Reserves

The roots absorb water and nutrients from the soil and move them up to the leaves, where CO_2_ is photosynthetically reduced to carbohydrates. Those carbohydrates not readily used by the plant’s cell metabolism are stored, mainly in leaves and branches, with a smaller amount being stored in the roots. The maximum amount of carbohydrates the tree stores is reached just before spring flush [16,17].

The straighter and more vertical the branch is, the more intense the xylem flow rate is [18]. High transpiration rates promote a flow rise in the xylem sap and increase photosynthetic rates, thus boosting carbohydrate synthesis and phloem upload. This favors the development of the tree’s straighter and more vertical branches and upper parts.

On the other hand, phloem sap movement is more difficult in horizontal and tortuous branches. Consequently, phloem sap retention and accumulation in the leaves occurs, promoting flower induction and fruiting [19]. This principle is the basis for some complementary operations to pruning [20] such as branch girdling and bending. Branch girdling is often practiced to enhance fruit set, fruit size, and citrus trees’ yield, especially in mandarins [21]. Based on the same principle, branch bending stimulates the formation of a larger number of flowers and fruits [19,22]

As previously seen, flower shoots arise from the previous year’s shoots (Figure 3). As new growth builds over previous growth, productive branches are increasingly located on the outer part of the canopy. Over time, this increases the tree size, promoting denser foliage outside the canopy that shades the inside. This way, fruits become located only on the outside of the tree canopy, where they are more exposed to wind and sun damage [23].

Cutting off part of the plant (e.g., a branch) removes some reserves [24,25], but the sap that would go to the removed part is redirected to the remaining parts, increasing the plant’s vegetative vigor. Nevertheless, pruned branches tend to be weakened compared to unpruned ones [26].

### 2.4. Types of Branches

According to their function, branches can be classified into (i) structural and (ii) production branches [27].

Structural branches are the largest in diameter, include scaffold branches and secondary and tertiary limbs, and form, with the trunk, the skeleton of the tree, determining its shape [5,27]. These branches do not produce fruits.

Production branches are small-diameter branches and twigs growing from the structural branches. Compared to the structural branches, they have relatively horizontal growth [27]. The fruits grow from these branches or their offshoots.

### 2.5. Tree Growth Habit

A citrus tree is a low-head tree in which scaffold branches arise from the trunk within a 50–60 cm height from the ground level. The branch insertion (crotch) angle with the main trunk differs according to the growth habit of each cultivar. Based on this tendency, citrus cultivars can be classified into three different groups: (a) upright, (b) spreading, and (c) drooping growth habit (Figure 5).

Upright cultivars (Figure 5a) form tall vertical branches with acute (narrow) insertion angles. These cultivars tend to form tall canopies. Examples are ‘Marisol’, ‘Salustiana’, and ‘Afourer’ (a.k.a. ‘Nadorcott’) cultivars.

Spreading cultivars (Figure 5b) form lower canopies with less acute branch insertion angles than upright ones. Examples are the “navel” and “blood” orange tree groups, ‘Valencia Late’ orange, and ‘Marsh’ grapefruit.

Drooping cultivars (Figure 5c) form horizontal branches with wider (open) insertion angles, resulting in lower canopies than in upright and spreading cultivars. Examples are the mandarins from the satsuma group, ‘Clemenules’, ‘Fortune’, and ‘Oroblanco’.

### 2.6. Apical Dominance and Hormonal Relations

Apical dominance refers to the control that the shoot’s terminal bud exerts over lateral bud growth and explains many characteristics of tree growth and response to pruning [28].

A branch has two types of buds: (i) the terminal bud, located at the tip of the branch, that controls the longitudinal growth of the branch, and (ii) several lateral buds located in the leaf axils and distributed along the length of the branch. The apical meristem of a branch produces auxins, which migrate downwards, suppressing the new growth of axillary buds [29]. This way, while a branch elongates, no lateral shoots appear (except under special conditions). Removing the apex of a branch leads to the breakdown of apical dominance due to the elimination of the auxin source, and new lateral growth can occur as the axillary buds are no longer suppressed [30].

If a branch stops growing without terminal bud removal, a new shoot will emerge from the terminal bud at one of the following shoot growth flushes. New side shoots may or may not appear just below (depending on ambient conditions and shooting habits). Some citrus species, such as mandarin trees of the satsuma group, have strong apical dominance.

### 2.7. Alternate Bearing

Alternate bearing can occur in some cultivars and refers to a cyclical yield in which a high yield with many small-sized fruits in one year (on-year) alternates with a low yield with few big-sized fruits the following year (off-year) [31] (Figure 6).

In alternate-bearing cultivars, a high fruit load during flower induction suppresses the flower-promoting genes [31,32,33,34]. Moreover, fruits may inhibit budbreak and, because fruit formation is highly resource-demanding, reserve depletion is high, and carbohydrate availability becomes scarce after an on-year. Consequently, the high fruit load on the tree in an on-year negatively affects the formation of vegetative shoots in summer and autumn, leading to poor flowering in the following spring and a high fruit set [35,36,37]. This results in an off-year with a meagre or almost non-existent yield wherein competition between fruits is low and reserves are consumed only to a small extent [35].

After an off-year, the following year’s flower induction is high. Furthermore, after a low yield, the reserve accumulation and the new shoot formation levels are high, leading to abundant flowering. Under these conditions, resource competition is intense, but the fruit set rate is also high, leading again to an on-year [35,36,37].

## 3. Brief Classification and Characterization of Citrus Cultivars

Citrus include includes oranges, mandarins, grapefruits, lemons, limes, hybrids, and others, which are classified into different species, groups, and cultivars (Table 1). Although genetically very similar, citrus plants strongly differ in terms of canopy shape, shooting habit, bloom number per year, and fruit ripening period. It is essential to consider all these aspects when planning pruning.

Table 1 refers to the tendencies of the most common cultivars in the Mediterranean area. Because cultivar characteristics also depend on the rootstock, plant age, edaphoclimatic conditions, and cultural practices, including pruning, many cultivars’ growth and shooting habits may not always correspond to Table 1 information.

Regarding the trees’ growth habits, some upright cultivars form vertical, long branches that eventually hang down and take horizontal or drooping positions. Therefore, they may appear to assume different shapes (e.g., ‘Afourer’). The tree growth habit also depends on the clone’s origin, e.g., among the numerous ‘Tarocco’ clones, those of nucellar origin have a more vigorous growth habit than those obtained in vitro via shoot-tip grafting. This requires pruning to be different in trees from different clones. With regards to bearing habit, the classification of cultivars is also only indicative since yield depends on many factors, and some cultivars can even constitute a fourth category—unproductive [37].

## 4. Pruning Fundamentals

### 4.1. Types of Cuts

There are three types of pruning cuts, depending on the cutting point (Figure 7): (a) heading cuts; (b) reduction or drop-crotch cuts, and (c) thinning cuts.

A heading cut (Figure 7a) removes the terminal portion of a branch. This way, apical dominance is broken, and branching occurs from the nodes below the cutting. This cut may stimulate branching at a particular place and reduces the sizes of very long branches.

Drop-crotch cuts (Figure 7b) involve cutting a branch back to a lateral branch. A drop-crotch cut aims to stimulate the tree to grow in a particular direction. The lateral branch should be at least 1/3 the size of the cut branch. Otherwise, the cut will promote the development of many water sprouts.

Thinning cuts (Figure 7c) remove the entire branch. They are usually made to reduce competition between too-close branches, remove a branch crossing, and/or rub more desirable branches. Thinning cuts are also used to remove undesirable and bad crotch-angled branches (usually where the angle is less than 45°), where the bark of the two branches often grows down and presses the branches, bark included, instead of forming a small ridge (branch–bark ridge).

### 4.2. Cutting Operation

#### 4.2.1. Cutting Techniques and Procedures

It is essential to consider the orientation of the cut as it affects the cut surface covering. A cut involving the removal of a branch inserted in a larger branch (thinning cut) should not be parallel to it because this will result in serious injury, as the cut will also eliminate part of the larger branch.

The branch–bark ridge and collar must be identified before the cut (Figure 8). The branch collar is where the larger branch grows with and around the lateral branch; usually, it is a slightly swollen zone at the base of the lateral branch. The cut should be made next to the branch collar without damaging it (Figure 8). Leaving a stub above the branch collar must be avoided as the stub will die and the cut surface will not be covered by bark tissue.

If the bottom of the branch collar is difficult to identify, the cut angle can be estimated, as illustrated in Figure 8. This is accomplished by visualizing an imaginary line extending from the lower edge of the largest branch. This line forms an angle α with the branch–bark ridge. The final cutting line must form an angle β with the imaginary line, which, in citrus, must be equal to or slightly higher than the angle α.

Thick branches require three cuts for their complete removal to prevent trunk bark tearing (Figure 8). The first cut is made from the underside of the branch, about 40 cm away from the larger branch. The cut should be as deep as possible before the weight of the branch blocks the saw. The second cut is made further forward than the first cut, from top to bottom, so that the branch breaks between the two cuts without tearing the bark. After the second cut, there remains a stub, which is removed by the third cut (final cut line), which should begin on the outside of the tree bark ridge and end just outside of the branch collar, visible by the swelling at the bottom of the branch [38].

A reduction or drop-crotch cut must be made above a lateral branch capable of replacing the eliminated branch. For this to be possible, the lateral branch must be at least one-third the diameter of the eliminated branch. A reduction cut should bisect the angle formed by the branch–bark ridge and an imaginary line perpendicular to the stem to be eliminated.

#### 4.2.2. Wound Sealing and Protection

The plant tissues naturally seal wounds of small diameter by forming a healing periderm. The larger the wound diameter is, the more difficult it is to cover. Therefore, for wounds with a larger diameter, a healing wax is recommended [39].

## 5. Pruning Objectives in Citrus

In citriculture, pruning is performed with several different objectives (Figure 9).

### 5.1. Control of Tree Development and Shape

Unmanaged citrus trees develop irregularly. Most tend to form upright, long branches that split at the tip and form dense foliage outside the canopy. Branches form chaotically, often being intertwined and very close to each other [5,40]. These “wild” characteristics of citrus growth may present some limitations to cultivation. They hinder or prevent aeration and sunlight entry into the canopy, photosynthesis becomes limited, and many branches eventually die, becoming dry and making a large part of the canopy unproductive. On the other hand, controlling the height of the canopy is essential to avoid fruit forming on branches that are too high, preventing or minimizing the need to use ladders for harvesting, making harvesting safer and more efficient.

Yield does not increase proportionally with an increasing canopy size. Oversized canopies may even result in lower yields [41]. The rootstock choice is essential as it decisively affects the canopy’s development and shape. Choosing a suitable rootstock will reduce the need for and cost of pruning [42].

In short, the control of tree development and shape allows for the following:Shaping the tree’s structure, which will be the physical support for the fruits; this is achieved through formative pruning.Maintaining the tree’s shape and preserving the training system.Controlling the canopy size.Controlling the production branches’ distribution to allow a good light exposition.Leaving enough floral buds to ensure optimal production.

### 5.2. Increase in Fruit Quality and Size

Removing dry and unproductive branches and promoting a good distribution of production branches allow for better nutrient distribution and use, better aeration, and better light distribution in the canopy [5,43,44,45,46]. Usually, fruits grown inside the canopy are of higher quality [41,46].

Frequently, high fruit-number-to-foliar-area ratios result in smaller fruits. Thus, removing part of the set fruits or preventing their formation by suppressing branches before or immediately after flowering limits fruit formation and allows the remaining fruits to grow bigger [46,47]. Removing less vigorous branches stimulates the development of better-formed branches capable of producing larger fruits [5,44,48,49]. Also, fruits formed at the ends of long, curved, or weak branches are smaller and of lower quality than those formed closer to the tree’s skeleton [17].

### 5.3. Alternate Bearing Management

As previously mentioned, alternate bearing is a cyclical production pattern in which an on-year alternates with an off-year. The small size of the fruits in on-years is undesirable as these fruits are less well accepted by the market [50] and, for example, in mandarin cultivars, fruit size is decisive to profitability [51].

Alternate bearing can be mitigated by pruning after harvest [46], with low-intensity pruning in low flowering years and high-intensity pruning in high flowering years [52,53,54,55].

### 5.4. Pest and Disease Control

Removing diseased and infested branches can be a direct form of pest and disease control [56,57,58,59]. Pruning alters the microclimatic conditions within the canopy, thus influencing the survivability of various pests [48,60] and their natural enemies [61]. Pruning also favors aeration and the entrance of solar radiation into the canopy, thus avoiding branch death, which can otherwise be a reservoir of microorganism resistance structures (e.g., fungal spores) that, under favorable conditions, provoke disease [62,63]. Phytosanitary treatments also become more efficient in pruned trees as the sprays better reach the inside canopy.

In *Citrus reticulata*, a normal intensity annual maintenance pruning (20% of canopy removal) can reduce fruit infection percentage for diseases such as scab, melanose, and canker [64]. Leaving pruned branches in the orchard can help the cultural controlling of some pests, such as the citrus leafminer, by reducing the number of live larvae and preserving some of its parasitoids [65]. Conversely, in citrus-growing countries where the tracheo-mycotic fungus *Plenodomus tracheiphilus* (causal agent of mal secco disease) is present, the pathogen must be controlled via a late-spring or summer pruning of the diseased twigs, which must be burned to reduce the inoculum [66].

Pruning may also have some disadvantages. For instance, in the short term, it increases sprouting and, consequently, aphid and other pest infestations [67]. Pruning can also stimulate the effects of nematode infection [58]. Although some works indicate that the pruning of symptomatic branches can control citrus variegated chlorosis [59], some authors consider it impossible to control wood diseases such as virus, bacteria, or fungi diseases through pruning [68,69,70,71,72,73,74,75].

### 5.5. Production Cost Reduction

Pruning represents a significant cost to the growers. However, if performed correctly and consistently, pruning will promote a better tree configuration that benefits other cultural operations [5] and compensate for its initial cost.

By making phytosanitary treatments more efficient, pruning reduces the number of necessary sprayings and the damage caused by some pests and diseases. Harvest also becomes easier and more comfortable since fruits become better distributed throughout the canopy, and the frequent use of ladders is no longer necessary since the canopy’s size is better controlled. Additionally, pruning itself becomes faster and easier in the following years [5,76].

## 6. Pruning Time

### 6.1. Young Trees

Formative pruning can be performed on young trees at any time if performed out of the frost period, preferably during spring [77]. In vigorous plants, unwanted young branches must be cut to avoid being removed as large branches in the future.

### 6.2. Mature Trees

In mature trees, pruning should be performed two to three weeks after fruit harvest [78], never during the frost danger period, and, if possible, before spring shooting to avoid interrupting the sap flow [5,79]. There are essentially three pruning periods, depending on the fruit development process, which is different depending on the cultivar (see Figure 1):

**First period:** It lasts from after harvest to flowering. It is the most suitable period for early-ripening cultivars (Table 1), which must be pruned before the spring shoot growth flush, thus avoiding sap circulation interruption. In these cultivars, early pruning favors the following harvest precocity [76,80,81]. However, it is essential to be aware of the frost risk, which will force the postponement of pruning [5].

When performed at this period, pruning has a positive effect on fruit set for two reasons: (i) by removing weak branches that produce numerous generative shoots (MFG and SFG shoots), it eliminates competition between generative organs; (ii) it increases the vigor of the spring shoot growth flush and thus the number of mixed shoots (MFM and SFM shoots), enhancing the probability of fruit set [17]. Pruning performed at this period can also improve fruit size [76].

**Second period:** It lasts from the time when the petals fall off to the end of the natural fruit (June) drop. It is the most adequate pruning period for alternate-bearing and mid-season cultivars (Table 1). Pruning can be performed a few weeks after harvest, as frost is no longer probable [76,80,81].

**Third period:** It lasts from the end of the natural fruit drop (June) to the end of August or later. Late-ripening cultivars (Table 1), mid-season cultivars when the harvest is delayed, and cultivars with production problems should be pruned during this period. Therefore, these cultivars can be pruned in early summer when vegetative activity stops due to hot weather [76].

Pruning in early autumn is not advisable as it can stimulate the formation of a later shoot growth flush that is more susceptible to frost damage in the winter [81].

## 7. Pruning Frequency

Pruning frequency depends on several factors such as species/cultivar, pruning requirements, pruning intensity, cost, labor availability, and orchard management [79]. It can be performed annually or every two, three, or more years. The less frequently pruning is performed, the more intense it must be when performed, and vice versa [5].

The longer the periods between pruning operations [82] are, the thicker the branches and the more time-demanding pruning will be. The cuts will be larger, causing larger wounds that will take longer to seal while providing entry for pathogens.

Annual pruning is the most appropriate for vigorous cultivars/trees while less frequent pruning may be sufficient for less vigorous cultivars/trees. Pruning may be necessary in very low-vigorous cultivars, to promote the development of productive branches [12]. Annual maintenance pruning is also required to control alternate bearing problems.

## 8. Pruning Intensity and Severity

Pruning intensity refers to the amount of vegetation removed from the tree per unit of canopy volume.

Pruning severity is defined at the branch scale. It refers to the length of the branch’s removed part relative to its original length. The higher the proportion of the removed portion is, the more severe the pruning is.

Pruning intensity and severity depend on the purpose, frequency, and whether the tree is young or adult [5,77].

### 8.1. Young Trees

Intense pruning should be avoided and is practically unjustified in young plants [77,83]. When carried out at such an early stage, intense pruning causes a strong imbalance in a plant and negatively affects the future equilibrium of the tree. On trees that have not yet produced, the entry into production is delayed, and in trees that have already begun to produce, production decreases and may even be temporarily interrupted [76,84].

The vegetation volume removed from young trees should not exceed 25% of the plant’s total volume [77].

### 8.2. Mature Trees

In mature trees, pruning intensity depends on the pruning type, frequency, and purpose (Table 2). When pruning is annual, it does not need to be intense [85]. The lower the frequency is, the more intense the pruning should be [5]. A very strong pruning only applies to specific cases, never to maintenance pruning.

The more severe the pruning of a branch is, the longer the shoots that emerge from it are. Although longer, these shoots do not necessarily have more nodes [9]. Additionally, the shoots that emerge from pruned branches are usually longer than those that arise from unpruned ones [26]. However, this does not mean that the new shoots’ growth is sufficient to compensate for the length of a branch removed by pruning [9,26]. If the objective is to control the formation of flowers or fruits, pruning intensity and severity should be appropriate; otherwise, pruning may be useless [86].

## 9. Pruning Types

Four types of pruning can be considered, depending on the stage of the tree’s life: (i) formative pruning; (ii) maintenance pruning; (iii) recovery pruning; and (iv) rejuvenation pruning. The central aspects of each type of pruning are summarized in Table 3.

### 9.1. Formative Pruning

Formative pruning is carried out during the first years of the tree’s life, from the nursery to when the plant reaches its final size. It aims to obtain the desired tree structure.

The formative pruning objectives are:To establish an adequate, healthy, robust branch structure that can support the future tree canopy and a good fruit load even under adverse weather conditions.To prevent production branches from forming too close to the ground.To optimize production branch distribution, avoiding competition for space and light.To perform the structural management of the future canopy to contain the minimum wood for its structure and the maximum leaf surface area exposed to solar light.

#### 9.1.1. Nursery and Planting Pruning

Nursery pruning (Figure 10) is performed after the shoot of the scion has developed [87]. The rootstock stem must be removed. If the scion is not equal to or thicker than the rootstock stem, this must be cut off, leaving a 3 to 5 cm stub, which will be eliminated after the scion thickens.

When the chosen training system is of the ‘free’, ‘traditional’, ‘dichotomic’, or ‘open-center’ kind, i.e., requires the formation of scaffold branches, the cutting of the main branch can be performed while in the nursery, between 0.6 and 0.7 m above the ground (heading cut). This allows the formation of new shoots, still in the nursery stage, which will develop into future scaffold branches (Figure 10a). As most plants sold by nurseries in Mediterranean countries have only the main stems (Figure 10b), a heading cut must be performed at planting (Figure 10c).

When the chosen training system is the ‘central leader’ or similar, the plant axis must be maintained in the nursery and after planting in the field.

#### 9.1.2. Pruning after Planting

Formative pruning continues in the years after planting, until the tree has the desired shape and size, and is carried out according to the chosen training system [87]. Regardless of the training system chosen, and even if the free training system is chosen, some aspects must be considered in formative pruning:The formation of the principal branches too low or right next to the grafting zone should be strongly avoided.The trunk should always be at least 0.5 m above the ground.The insertion angle of the principal branches with the trunk should not be too narrow or too open.

### 9.2. Maintenance or Fruiting Pruning

Maintenance pruning starts from the moment the tree enters full production. In the adult phase, a balance between vegetative and productive development should be promoted while also favoring productive branch renewal. At the same time, enabling aeration and light penetration into the canopy is crucial. Maintenance pruning also improves nutrient distribution across branches [43,78,85,88].

The main objectives of maintenance pruning are [44,89,90,91,92,93]:To control the vegetative growth.To promote the balance between vegetative and productive development.To promote the renewal of production branches.To maintain or improve the light distribution and air circulation inside the canopy.To maintain the training system.To regulate the production (alternate bearing management).To improve fruit size and quality.To improve fruit set and yield.

#### 9.2.1. General Considerations

In maintenance pruning, it is essential to consider the following aspects: (i) the vigor of the tree; (ii) branch and foliage density; (iii) the balance of the canopy; and (iv) skirt level [80].

**Vigor of the tree:** The less vigorous the trees are, the more intense pruning should be, and vice versa [43,80].

**Branch and foliage density:** The degree of branching and foliage density differs between cultivars, being mostly related to shooting habit. Short multiple shoot (SMS) cultivars tend to form dense and compact canopies, while long solitary shoot (LSS) cultivars tend to develop more scattered branches. Branch and foliage density must be controlled using thinning cuts [80] by removing weak, dry, diseased, or poorly positioned branches and water sprouts [43].

**Balance of the canopy:** Vertical branches are usually very vigorous, almost unproductive, and difficult to reach, so they must be pruned. Conversely, horizontal branches, which are more productive, should not be removed. In the case of horizontal branches, it is sufficient to control their development by cutting the longest ones, those that cross each other, and those that make the canopy too dense. Removing a branch facilitates the nutrition and illumination of the neighboring branches, improving their production [43,88].

**Skirts level:** Skirts are the branches in the lower part of a canopy, and their height must be controlled. Skirts that are too high reduce the fruiting surface, expose the trunk to the sun, favor weed development, and soil moisture evaporation. On the other hand, skirts that are too low make the fruit more susceptible to diseases caused by soil fungi and some pests and make some cultural practices, such as weed control, more difficult. A skirt’s level should be high enough that the fruits formed in this zone never come into contact with the ground [80,94].

The maintenance pruning frequency should be managed so that exceeding the normal pruning intensity (removal of 20% of the canopy) is unnecessary. This depends on several factors, such as the cultivar, rootstock, soil, nutrition, climate, etc., which determine the rate and type of tree growth [5,95].

In short, after maintenance pruning, the canopy of the tree should have (Figure 11) [95]:A top and a lateral opening to favor light entry, aeration, and phytosanitary treatments.A not-too-dense external surface of the canopy that will allow good aeration and uniform vegetation distribution.Not-too-low skirts.Production branches at a height that facilitates harvesting.

#### 9.2.2. Alternate Bearing Management

In alternate-bearing cultivars, a high fruit load in one year leads to the severe suppression of flowering the following year [34]. Therefore, in the year of abundant flowering, these cultivars should be 30%-intensity pruned [5,46,96]. Conversely, pruning should be light (10%) or not performed in the scarce-flowering year (Table 4).

### 9.3. Recovery Pruning

Recovery pruning is a strong-to-very-strong-intensity pruning carried out after several years without regular maintenance pruning [41,97]. It aims at renewing part of the canopy and simultaneously correcting problems related to oversized canopies (Figure 12) [41,73].

The fruit development period lasts for over one year in late cultivars such as the orange trees ‘Valencia Late’, ‘Ovale’, and ‘Dom João’. No matter when such a cultivar is pruned, pruning results in fruit removal, which directly reduces the subsequent yield. Therefore, many growers avoid pruning, postpone it, or prune too little to prevent production losses while also reducing the operation’s cost. When maintenance pruning is not performed or is improperly performed, trees tend to form large crowns with dense foliage on the peripheries (Figure 12a) [41,78].

There are several limitations related to oversized canopies: dense foliage outside limits the aeration and light entry through the canopy [41,89,98]; the lack of light causes branches to die, making the interior of the canopy an unproductive zone; deadwood can also serve as an inoculum for phytopathogenic fungi [99]; phytosanitary treatments become more difficult and less effective; and harvesting becomes less efficient, more dangerous, and more expensive due to the need to use ladders [41,98]. However, fruits developed inside the canopy are usually of higher quality [90,100].

Using recovery pruning, some scaffold and large branches should be removed with thinning cuts (Figure 12b) to open the canopy upward (Figure 12c) and thin it out in the peripheral areas. The recommended pruning intensity is between 30 and 50%, never exceeding 50% [41,82].

The checking of the branches to be removed must be done from the inside of the tree canopy, where it is possible to better evaluate branch structure and distribution. Recovery pruning is also an excellent opportunity to remove large-diameter or poorly positioned branches left by inadequate or poorly executed formative pruning [5,73].

Recovery pruning reduces the next year’s yield [97] in late-ripening cultivars such as the orange trees ‘Valencia Late’, ‘Ovale’, and ‘Dom João’. However, a few years after pruning, cumulative production may be higher on pruned trees, and the fruits of pruned trees tend to be larger and of better quality [41].

### 9.4. Rejuvenation Pruning

Rejuvenation pruning aims to rejuvenate trees and can be performed on aged and depleted yet healthy trees (Figure 13). It removes almost the entire tree crown, leaving the trunk and scaffold branches (Figure 13a). If the trunk and structural branches are not healthy due to scalds, diseases, or viruses, this type of pruning is not viable [71,78].

Although old, the trunk and scaffold branches contain adventitious buds from which new shoots can emerge. New branches can form from these new shoots, allowing canopy renewal (Figure 13b). After the formation of new shoots, a formative pruning is required to control the formation of the new crown of the tree (Figure 13c). Parallel to this type of pruning, another cultivar can be introduced via grafting [101].

Rejuvenation pruning causes a strong imbalance between the above- and below-ground parts of the plant [102]. Therefore, cultural practices must be adjusted after pruning to reduce the sap flow into the above-ground portion. While fertilization and irrigation should be limited and reduced, it may be necessary to intensify other practices, such as eliminating weeds, which are more prevalent in unshaded soil. After rejuvenation pruning, trunks, scaffold branches, and large branches that were previously shaded and covered by foliage become suddenly fully exposed to solar radiation. This exposure can cause severe burns, which must be avoided as much as possible. For this reason, trunks and branches exposed to the sun should be painted (Figure 13a). Applying a whitewash or watered-down white latex non-phytotoxic paint that acts as a sunscreen is recommended. It is important to note that the rootstock influences the recovery of the trees after this type of pruning [103].

### 9.5. Other Types of Pruning

There are other brief pruning operations that some growers may consider necessary, such as cleaning pruning, phytosanitary pruning, and water sprout removal.

**Cleaning pruning** removes dead branches affected by biotic or abiotic stress (frost, burns or phytotoxicity). It is a straightforward operation that does not require as much expertise as other pruning types. Deadwood removal can be performed at any time of the year [78,104].

**Phytosanitary pruning** removes diseased or pest-infected branches to control them or minimize their effects [59].

**Water sprout removal** should be performed right after the shoots have sprouted, when it can still be performed by hand and glove without pruning shears [104]. This practice is essential after maintenance pruning, when several water sprouts appear. It is a short operation that does not require as much skilled labor as other pruning types [100].

## 10. Training Systems

### 10.1. Free Training System

Free training constitutes the simplest and easiest training system, which allows the plant to grow freely, leading to a faster entry into production than other, stricter training systems. However, some interventions are necessary to guarantee a balanced, well-formed canopy and promote the good development of the tree. In this training system, suppressions are light and aim to remove rootstock shoots, water sprouts, and aged and poorly positioned branches [5,77]. It starts by removing the shoots that emerge from the rootstock [77,105,106]. In the adult stage, it is important to control the water sprouts [77].

The main purposes of the free training system are [77]:To respect the unrestrained growth of the tree as much as possible and achieve a greater canopy volume in a shorter time.To accelerate the entry of the tree into production.To save time and labor in pruning execution.

The free training system is suitable for the following cultivars: (i) fast-production entry cultivars such as ‘Lanelate’ or ‘Orogrande’; (ii) highly vigorous cultivars such as ‘Afourer’, ‘Fortune’, or ‘Ortanique’; (iii) slow-growing light-pruning-requiring cultivars such as ‘Navelate’, ‘Oronules’, or ‘Nova’.

### 10.2. Dichotomic Training System

The dichotomic training system is the strictest citrus training system. It is easy to implement and promotes the tree’s good structure, with an optimal distribution of structural branches. This training system arranges the structural branches according to their hierarchy and is based on successive dichotomies: two scaffold branches originate from the trunk, then two branches originate from each scaffold branch, and so on, until the structure of the tree canopy is formed (Figure 14).

The dichotomic training system begins with the cutting of the main branch (Figure 14a), which is performed at the nursery, or planting pruning (see Section 9.1) [5,105]. After this cut, the dichotomic structure of the tree is achieved via a series of several successive steps:

**First step** (Figure 14b)**:** Two of the several branches that emerge below the trunk cut are chosen as the main scaffold branches. These two scaffold branches should be inserted in opposite directions and at different heights (about 10 cm difference). They must be cut at around a 50 cm length to allow the next dichotomic level to form. All the other branches must be removed; if this operation exceeds the recommended pruning intensity for young plants, some branches may stay and be shortened to half their length. All the shoots emerging from the rootstock must be removed [5,106].

**Second step** (Figure 14c)**:** This must occur after new branches have emerged from the two scaffold branches selected and pruned in the first step. Using the same first-step criteria, two new branches must be chosen from each scaffold branch. The new branches will form the second dichotomic level and should be oriented perpendicularly to the previous dichotomic level. Except for the scaffold branches, the trunk branches that have remained from the first step or have originated since must now be removed (thinning cut) [105,106].

**Third step** (Figure 14d)**:** The new branches emerging from the second dichotomic branch level will form the third dichotomic level. The third dichotomic level’s branches must be cut about 50 cm above and perpendicular to the origin of the second branch level (see top view). Non-structural and vertical branches in the canopy’s interior must be removed. New branches close to the selected ones that may slow their development must be removed. Some branches that may be well positioned to fruit may be kept. If they are long and vigorous enough, they should be cut (purple lines) to break their apical dominance and turn them into temporary production branches [106].

**Fourth step** (Figure 14e)**:** The fourth dichotomic branch level is obtained following the same rationale as the previous steps. The previous step’s temporary production branches should be removed after fruiting and harvesting to avoid becoming too vigorous. At this stage, the branches kept in the previous step should be long enough to be cut and develop lateral branches for production. As in the third step, and if appropriately positioned, some of the non-selected new branches can be left and cut to become production branches.

The final structure of the tree must present the structural branches equitably distributed throughout the canopy. From the structural branches will emerge the production branches (Figure 14f) [5,106].

### 10.3. Traditional Training System

This system allows the formation of an open canopy, favoring its aeration and light exposure. It consists of choosing three or four scaffold branches from which the rest of the canopy structure will emerge. Three or four vigorous and evenly oriented branches are selected from those originating below the nursery or planting pruning point. If three branches are chosen, they should make an angle of approximately 120 degrees; this angle must be 90 degrees if four branches are selected. The three or four chosen scaffold branches must not be too low or must be at the same height and should have an intermediate inclination that is neither too vertical nor too horizontal. Any branch emerging from the trunk other than those selected as scaffold branches should be removed.

The shoots emerging from the scaffold branches and oriented to the canopy’s middle can be cut halfway and later removed.

From each scaffold branch, two or three evenly distributed secondary branches must be chosen to continue the tree structure. The selected secondary branches must also face the canopy exterior to favor its spreading.

From these secondary branches, tertiary branches will emerge. Two or three tertiary branches will then be chosen using the same criteria as before. This step is repeated in subsequent years until the final size of the tree canopy is reached. Branches threatening to alter the tree’s structure must be eliminated during maintenance pruning.

### 10.4. Central Leader Training System

The central leader training system gives the tree a pyramidal shape (Figure 15) [107]. This training system can only be chosen if the grower intends to establish a high-density orchard (Figure 15a) [107,108,109], and is best suited for upright-crown-shape cultivars (Table 1) such as lemons (1 m × 3 m) [107] and ‘Afourer’ mandarin (1.5 m × 5 m) [108].

A vertical axis forms the trunk of this training system. The tree axis should be dominant over the side branches (Figure 15), which should have a large angle of incidence (Figure 15b), making them well connected and not easily breakable [107].

During the growth period, the formation of side shoots should be stimulated to establish side branches at the desired levels of the tree structure, for which making a heading cut on the main branch may be necessary. New side shoots’ growth must be controlled, and a shoot must be selected to replace the axis.

It is important that side shoots form evenly spaced around the axis. This can be achieved by selecting some horizontal side shoots formed in response to the branch cut. Vertically growing branches that are not part of the tree structure must be removed. Vertical shoots in suitable positions to become productive branches can be bent to a more appropriate angle rather than removed [107].

### 10.5. Vase or Open-Center Training System

The vase or open-center training system (Figure 16) is often applied to some stone fruits, such as peaches and nectarines, and rarely to citrus fruits. However, this system optimizes aeration and sunlight distribution, which are very important in citrus plants. Four or five scaffold branches form the canopy structure in this training system.

In the second or third year after planting, the four or five most vigorous and better-oriented shoots must be selected to be the future scaffold branches. The chosen branches should be roughly radially equiangular, not be too low or at the same height, and have an intermediate inclination, neither too vertical nor too horizontal.

Once the main branches have been selected, the secondary branches directed towards the crown’s inside must be removed. A drop-crotch cut should be performed at the top of the main branches over one of the most vigorous side shoots to encourage side-branch growth and canopy opening (Figure 16a).

When the main branches reach the desired height, they should be pruned over a weak side branch so that they lose vigor. All branches must be kept at the same height over the years (Figure 16b). This training system is widely used for Corsican clementines.

## 11. Mechanical Pruning

Manual pruning is one of the most expensive cultural practices in citrus production, and qualified pruner availability is scarce. Thus, mechanical pruning is an alternative, as it can significantly reduce the pruning cost [110,111,112]. However, mechanical pruning has some disadvantages compared to manual pruning (Table 5).

If the grower intends to fully mechanize pruning in the orchard, controlling a tree’s shape and size will be achieved differently from how it is in manual pruning. In an orchard where mechanical pruning is exclusively used, the trees have a free structure.

In mechanical pruning, a tree can be cut at the top, sides facing the inter-rows, and bottom. No cuts are made on the branches that develop towards the other trees in the row. This way, a hedge is formed in which the productive zone is the outer canopy layer facing the inter-rows where dense foliage and fruit develop. The canopy’s shape maintenance is accomplished using three cutting operations: (i) topping, (ii) hedging, and (iii) skirting (Figure 17) [83,94,113]. These cutting operations can also be used as complements to manual pruning.

### 11.1. Cutting Operations

#### 11.1.1. Hedging

Hedging consists of cutting the trees’ sides along the row, either vertically or at a slight angle (Figure 17). The numerous cut branches that result from hedging lead to several shoots, forming a foliage “wall” [83,113]. The space between canopies should be wide enough (2.0 m to 2.5 m) to allow machinery and equipment movement and give the trees access to sunlight [113].

Mechanical pruning cuts must be started before the inter-rows are closed and shading becomes a problem. In this way, the cuts are smaller and have the least possible effect on fruiting. The smaller the tree spacing is and the more vigorous the trees are, the earlier the first intervention is required and the higher the frequency of the interventions needs to be [83,113].

Hedging is generally performed at such an angle that the canopy top is pruned to a smaller width while the canopy bottom is cut to a larger width. The trimming angles lie between 0° and 25° (with an optimum between 10° and 15°) (Figure 17) and should optimize the trees’ solar radiation exposure [83,94,113,114,115].

Wider cutting angles allow for more sunlight reaching the lateral surfaces of trees, slower growth in the lower parts, and more vigorous growth in the upper part of the tree crowns. Additionally, they promote more effective phytosanitary treatments and improve harvesting efficiency. However, wider cutting angles reduce initial yield and stimulate branch-length growth, which can increase cold-damage susceptibility [113].

In the navel-group orange trees, hedging can be performed for bloom control in years of heavy flowering, where it limits fruit set and increases fruit size. When performed during the spring, hedging causes the summer shoot growth flush (vegetative shoots only) to be more synchronous. The following year’s generative and mixed shoots emerge from these summer-formed shoots (Figure 3). In the following spring, the trees will produce a higher percentage of mixed shoots and a lower percentage of generative shoots than trees where hedging has not been performed. Usually, mixed shoots produce larger fruits with a higher set rate than generative shoots [54,116].

In years of abundant flowering, hedging during bloom can be a strategy for alternate bearing management by (i) reducing the number of set fruits due to the reduction in flower number, thus decreasing the number of fruits in the on-year, and (ii) stimulating the synchronized and strong formation of vegetative shoots in the summer shoot growth flush, which begets a better fruiting capacity for the following year (off-year) [116].

If mechanical pruning is performed regularly and consistently, the production tends to remain the same over the years [113]. Conversely, if pruning is performed less frequently than is appropriate, excessive vegetative growth may happen, and thus, the removal of a relatively large portion of the canopy will become necessary, and a sharp decline in subsequent production will occur. In addition, more severe pruning is more expensive, and equipment wear is more problematic and costly [83,113].

#### 11.1.2. Topping

The topping consists of cutting off the tree canopy’s upper part and helps control the tree’s height (Figure 17) [83,113].

Topping should be performed before the trees in question get too tall and should be included in the orchard maintenance pruning program. When the canopy is too high, harvesting and phytopharmaceutical treatments are more difficult [83,113]. Lower cuts (stronger pruning intensity) result in the formation of longer shoots, and higher cuts (lighter pruning intensity) result in the formation of shorter shoots [9].

Longer intervals between topping operations are more expensive because the cut is harder, and wear and tear on cutting equipment is higher. Yield losses due to low-intensity topping are insignificant unless the trees are very large, especially since fruit density tends to be higher in smaller trees [113].

Topping should be performed in a flat cut (0° with the horizontal plane) whenever trees are small, narrow, or were hedged in an open angle. If the hedging angle is sufficient in narrow rows, topping can be performed in one pass down the row [113,114,117]. However, topping cut angles may also vary from 15° to 30° to the horizontal plane. In this case, the cutting process is easier, and the top of the tree crown becomes higher in the middle than on the sides (Figure 17) [83,94,113,117].

The optimal tree height depends on the distance between the trees, the hedging cutting angle, and the trees’ width. The cutting height can range from 3 m to 6 m. Commonly, low-cutting heights are done to increase fruit size or to renew declining trees.

Topping stimulates the formation of new shoots [118].

#### 11.1.3. Skirting

Tree skirts consist of the lower branches, which are closer to the ground (Figure 17). In some cultivars, skirts’ contributions to yield are significant. However, at that zone of a tree, the fruits become too close to the ground and are thus more susceptible to diseases caused by soil organisms (fungi and others), especially *Phytophthora*, which causes a high rate of fruit rot. Moreover, when the skirts are too close to the ground, their branches and fruits are often damaged by passing machinery, herbicide application, and other cultural practices.

Skirting is a pruning practice that eliminates the skirt’s lower part.

Skirting prevents the foliage and fruits from being too close to the ground [83,113], thus reducing the occurrence of fruit rot associated with soil organisms; facilitates the machinery and equipment movement and the herbicide applications; and allows a more efficient irrigation-system control [113,119,120]. This cutting operation may be very helpful for canopy management in cultivars with a drooping tree growth habit (e.g., *C. unshiu*) [121].

### 11.2. Pruning Program

If mechanical pruning operations are frequent, each vegetation removal is light; additionally, frequent interventions avoid large diameter cuts, minimizing wear on pruning blades [113]. The amount of biomass removed is higher when the pruning frequency is lower [111].

When pruning is performed mechanically in an orchard, it is essential to establish a pruning program (Table 6). This plan should include the frequency, the cutting operation, and parameters, considering the desired size and shape of the trees. Pruning programs may vary considerably depending on the cultivar, tree vigor, spacing, and grower’s preference [113].

The seven mechanical pruning strategies examples listed in Table 6 and their effects are summarized below (as mentioned before, no cuts are made on the branches that develop towards other trees in a row; thus, only the two canopy sides—Side 1 and Side 2—facing the inter-rows are pruned):

Strategy 1 (THH/THH) consists of completely pruning all sides of the tree each year. This strategy can be a good option for early, vigorous cultivars. Early cultivars can be pruned before spring bloom, preventing flower or fruit removal and stimulating spring bloom. Vigorous cultivars may benefit from annual pruning because less vegetation is removed, cuts are smaller, and the wear on pruning equipment is less than with a lower pruning frequency. In one study, this pruning strategy did not reduce ‘Navel Foyos’ orange tree production, and the costs were 82% lower than those of annual pruning performed exclusively by hand [122]. For the ‘Clemenules’ clementine, strategy 1 was the most expensive of the mechanized pruning strategies. Still, it was 34% cheaper than manual pruning, with no differences in yield or fruit size [111]. In ‘Star Ruby’ grapefruit, strategy 1 originated a higher yield than strategies 6 and 7 [123].

Strategy 2 (THH/--) consists of completely pruning all sides of the tree in alternate years. Strategy 2 can be a solution for less vigorous cultivars whose growth does not need to be controlled as much. In a study, this pruning strategy did not change yield, fruit size, and fruit quality in ‘Washington Navel’ orange compared to only manual or mixed (mechanical–manual) pruning. On the other hand, ‘Salustiana’ orange yield decreased by 17% compared to manual pruning, but there were no differences in fruit size and quality (average of 4 years of results) [124].

Strategy 3 (TH/TH) consists of pruning each side of the tree in alternate years while pruning the top annually. This strategy may be appropriate for cultivars that grow vertically or produce vigorous vertical branches and where control over vertical growth is more urgent than control over side growth. This strategy was the fastest, the cheapest, and the most productive for the ‘Fino 95′ lemon in a study [110]. This strategy was also the cheapest and fastest for the ‘Navel Foyos’ orange, costing 23% less than strategy 1 and 98% less than manual pruning; no differences were found in yield or fruit size [122]. For the ‘Clemenules’ clementine, strategy 3 was 62% cheaper than strategy 1, with no differences in yield or fruit size [111].

Strategy 4 (HH/T) consists of cutting both sides of the canopy or topping in alternate years. This strategy may be appropriate for cultivars of medium vigor, where the annual pruning of the entire canopy is not warranted.

Strategy 5 (H/H/T) consists of alternately pruning each side of the canopy or the top every three years. This strategy can be used for low-growing cultivars where more frequent pruning is unnecessary.

Strategy 6 (HH/HH) consists of annually pruning both sides of the canopy. It may be appropriate for vigorous, open-growing (strongly spreading or drooping) cultivars where topping is unnecessary. This strategy was 56% cheaper than strategy 1 and 88% cheaper than manual pruning (4-year average) for the ‘Clemenules’ clementine. No differences in fruit size or yield (4-year average) were observed compared to strategy 1 or manual pruning [111].

Strategy 7 (T/T) consists of topping every year. It may be appropriate for vigorous, upright-growing cultivars. In a study, this strategy led to the formation of larger fruits in the grapefruit ‘Star Ruby’ [123].

### 11.3. Mixed Mechanical-Manual Pruning Strategies

Mechanical pruning can be used as a complement to manual pruning. There are many different possible strategies, and it is up to the farmer to define a program that considers manual and mechanical pruning operations and their frequency [117].

One of the most frequent strategies is to mechanically prune the skirts (skirting) while the rest of the canopy is pruned manually. Another possible strategy is controlling the trees’ height using a regular topping operation while the middle part of the canopy is pruned manually [82]. Other growers may do a fully mechanical pruning one year and a manual pruning the next year [125] or a light manual pruning operation immediately after the mechanical pruning [110,111,117,122,125].

## 12. Pruning Waste Management

Depending on its intensity, pruning may produce significant waste (wood, leaves, and sometimes fruits) [126,127]. The disposal of pruning waste [128] is achieved in different ways. Some of them are summarized in the following subsections.

### 12.1. Shredding and Soil Incorporation

This practice consists of leaving the pruning waste in the orchard inter-rows to be mechanically shredded (Figure 18a). After shredding, the biomass is left on the surface, forming a layer of mulch (Figure 18b) [129]. Mulching has several benefits such as containing soil erosion (inter-rows are the areas where erosion is most pronounced) [129,130], reducing soil moisture evaporation, reducing surface runoff, and increasing water infiltration [130]. Shredding also allows the mechanical controlling of weeds [131]. Whenever there are pathological problems such as mal secco disease, pruning-waste incorporation is forbidden and this waste must be burned [66].

The crushed waste decomposes, promoting soil enrichment with organic matter [132]. This increases soil fertility and orchard productivity [133], improves the soil structure due to the agglomeration of mineral particles, and ameliorates soil porosity and permeability [134]. Water infiltration is also enhanced, avoiding erosion and decreasing evaporation [135] and thus improving the soil–water balance [133,134].

### 12.2. Other Practices

Waste incineration in the orchard is an outdated practice that must be avoided and is justified only in exceptional cases. Burning waste releases large amounts of carbon dioxide (CO_2_), methane (CH_4_), carbon monoxide (CO), and other gases, as well as tiny particles (smoke), into the atmosphere [136,137]. Therefore, this is an environmentally unsustainable practice [138], banned in some Mediterranean countries [139].

There are other uses for pruning waste that are less common but still possible [140,141]. Leaves can be used to extract essential oils by means of distillation, and the distilled leaf matter (bagasse) can be further used for animal feed. Woody material can be dried and then ground to produce pellets used in bedding for cattle (high absorption) or as biofuel [142].

## 13. Final Remarks

The objectives of citrus pruning are to control tree development and shape, increase fruit quality and yield, facilitate disease and pest control, reduce production costs, and mitigate alternate bearing. Understanding the general aspects of citrus morphology and physiology is crucial for designing appropriate pruning actions. Training systems and pruning procedures, techniques, and intensity must be carefully selected and executed considering the tree, edaphoclimatic conditions, and pruning objectives in question. Mechanical pruning is an alternative to strictly manual pruning as it can significantly reduce the pruning costs. Still, it has some disadvantages compared to manual pruning such as tree and fruit damage that negatively impacts the fruit production for the fresh market as a part of Mediterranean citriculture. Pruning generates waste that must be disposed of properly. Conversely, pruning waste may be a resource for soil enrichment and is valued as a raw material for essential oils, animal feed production, and other uses. Pruning affects a plant’s physiological behavior and is crucial for managing citrus trees’ growth, development, and production. Well-planned and well-executed pruning is essential for the prosperity of a citrus orchard.

## Figures and Tables

**Figure 1 plants-12-03360-f001:**
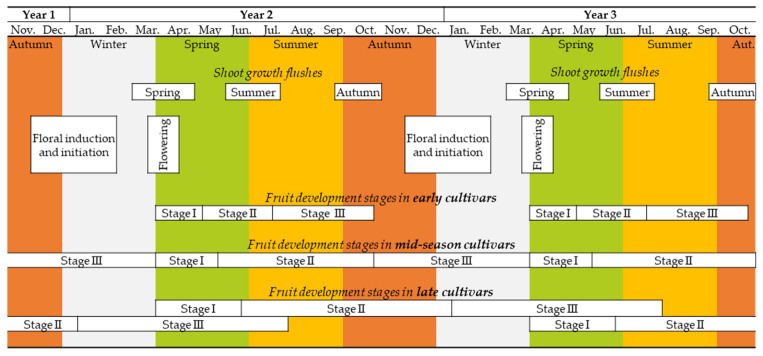
Summarized citrus phenological cycle in subtropical climates (Stages I, II, and III refer to the different fruit development stages; see Section 2.1.2. for details).

**Figure 2 plants-12-03360-f002:**
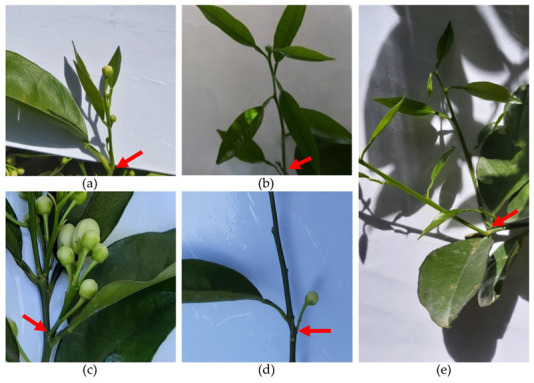
Types of shoots in citrus: (**a**) Multi-flower mixed shoot (MFM); (**b**) single-flower mixed shoot (SFM); (**c**) multi-flower generative shoot (MFG); (**d**) single-flower generative shoot (SFG); (**e**) vegetative shoot (V). The arrows mark the shoots’ insertions.

**Figure 3 plants-12-03360-f003:**
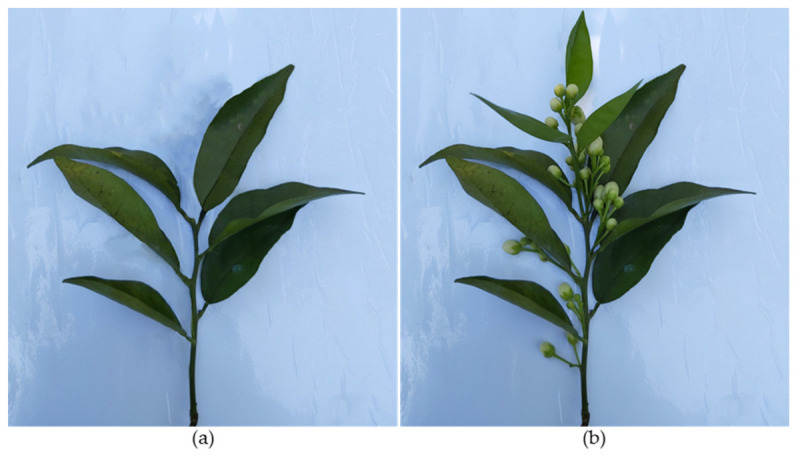
Shoots formed in different shoot growth flushes: (**a**) Vegetative shoot formed in autumn and (**b**) several generative and mixed shoots formed during spring from the axillary buds of shoot (**a**).

**Figure 4 plants-12-03360-f004:**
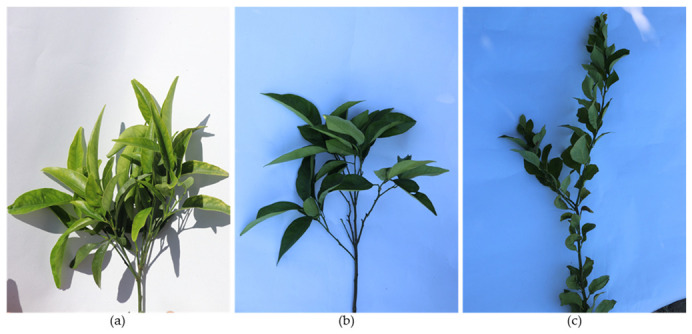
Shoots from trees of different shooting habit types: (**a**) SMSs, short multiple shoots (*Citrus clementina*); (**b**) IS, intermediate shooting (*Citrus sinensis*); (**c**) LSSs, long solitary shoots (*Citrus limon*).

**Figure 5 plants-12-03360-f005:**
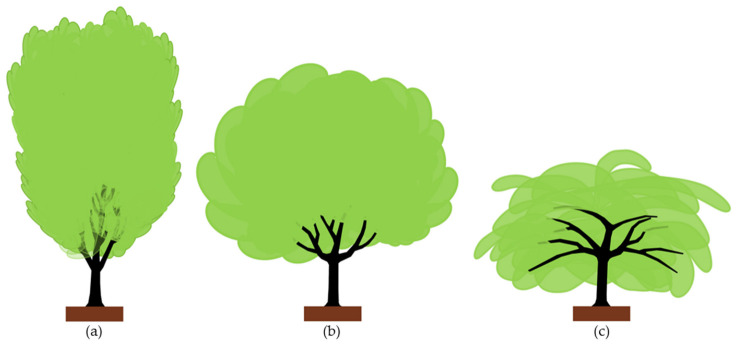
Types of tree growth habit in citrus: (**a**) Upright; (**b**) spreading; and (**c**) drooping.

**Figure 6 plants-12-03360-f006:**
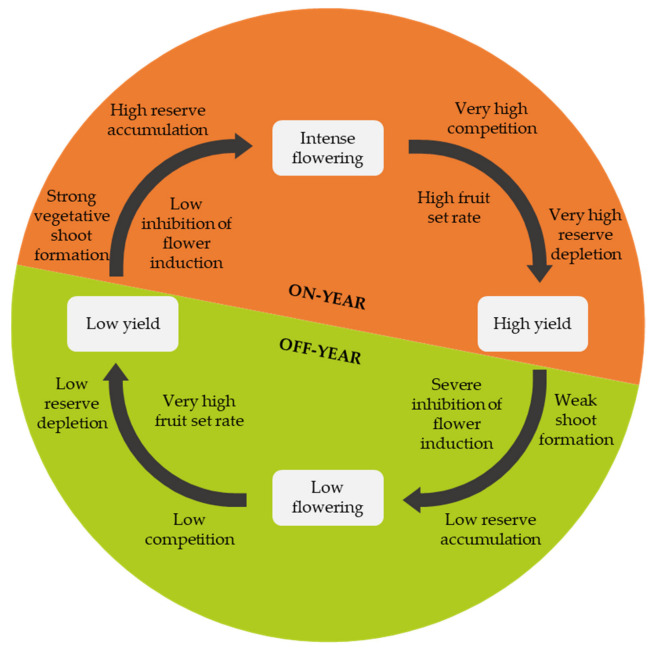
Citrus alternate bearing cycle.

**Figure 7 plants-12-03360-f007:**
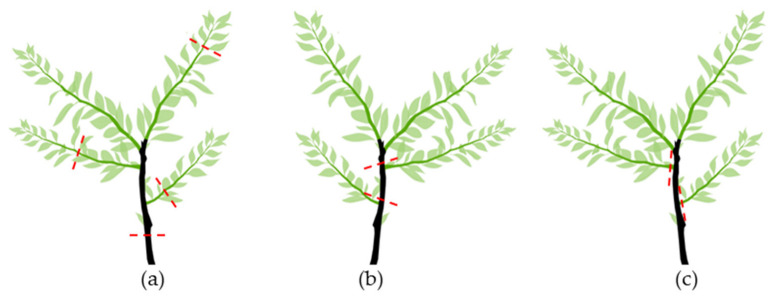
Types of cuts: (**a**) Heading cuts; (**b**) reduction or drop-crotch cuts; and (**c**) thinning cuts. The red lines mark the place where the cut is done.

**Figure 8 plants-12-03360-f008:**
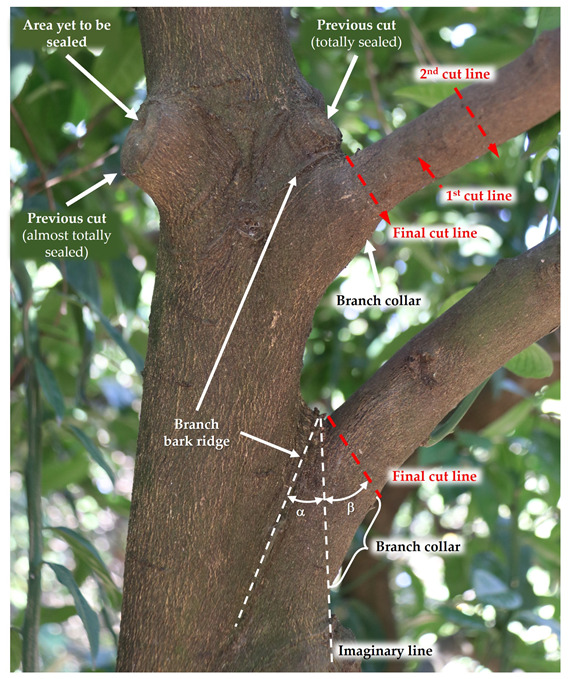
Cutting techniques.

**Figure 9 plants-12-03360-f009:**
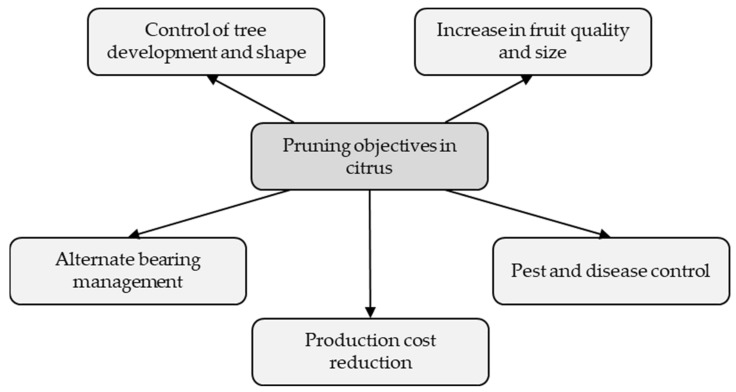
Main objectives of citrus pruning.

**Figure 10 plants-12-03360-f010:**
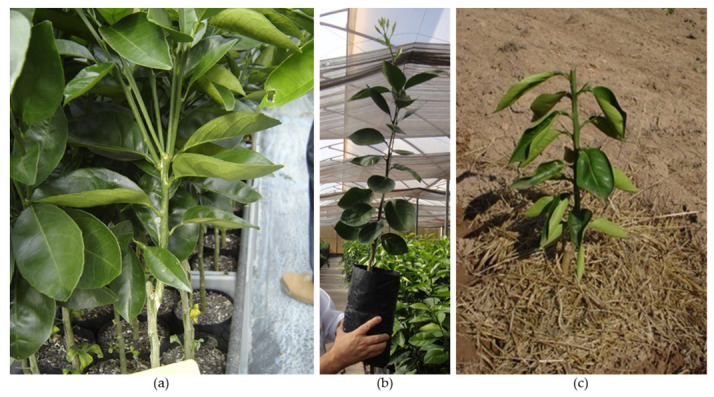
Nursery and planting pruning. (**a**) The heading cut of the main stem has already been performed in the nursery, allowing the formation of new branches. (**b**) The main stem is maintained in the nursery stage to allow the central leader formation, or it must be pruned at planting. (**c**) The heading cut of the main stem is made at planting, in the training systems that require it, if it has not been performed in the nursery.

**Figure 11 plants-12-03360-f011:**
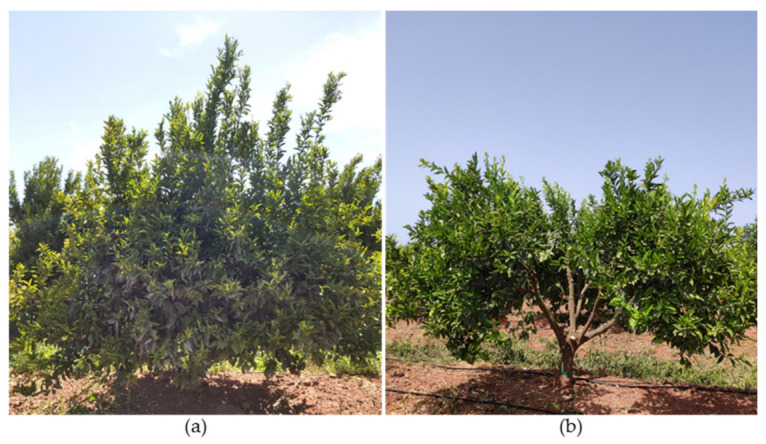
Annual maintenance pruning in ‘Encore’ mandarin. (**a**) Tree before pruning, with low skirts, relatively upright and with dense foliage, and too-tall branches in the center; (**b**) tree after pruning, with a lateral and a top opening, more scattered foliage, and higher skirts.

**Figure 12 plants-12-03360-f012:**
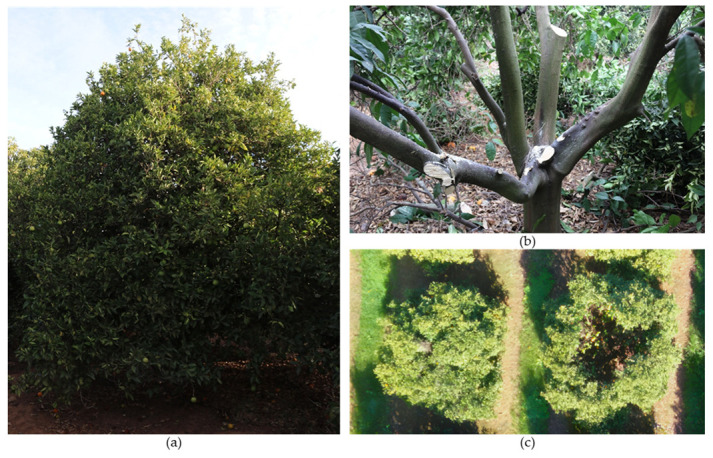
Recovery pruning can be a solution to correct several problems related to oversized canopies. (**a**) Oversized canopy in a ‘Valencia Late’ tree. (**b**) Large cuts for recovery pruning, resulting from the removal of large branches. (**c**) Comparison between a non-pruned tree (left) and a recovery-pruned tree (right) with an opening at the top of the canopy. (Photograph (**c**) by Carlos Guerrero).

**Figure 13 plants-12-03360-f013:**
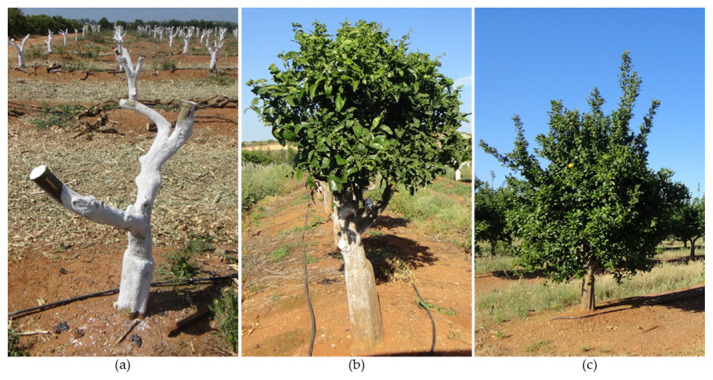
Rejuvenation pruning: (**a**) One month after pruning, with the introduction of another cultivar via grafting. Trunk and scaffold branches were painted to prevent solar burns. (**b**) New cultivar growth one year after pruning. (**c**) New cultivar growth two years after pruning.

**Figure 14 plants-12-03360-f014:**
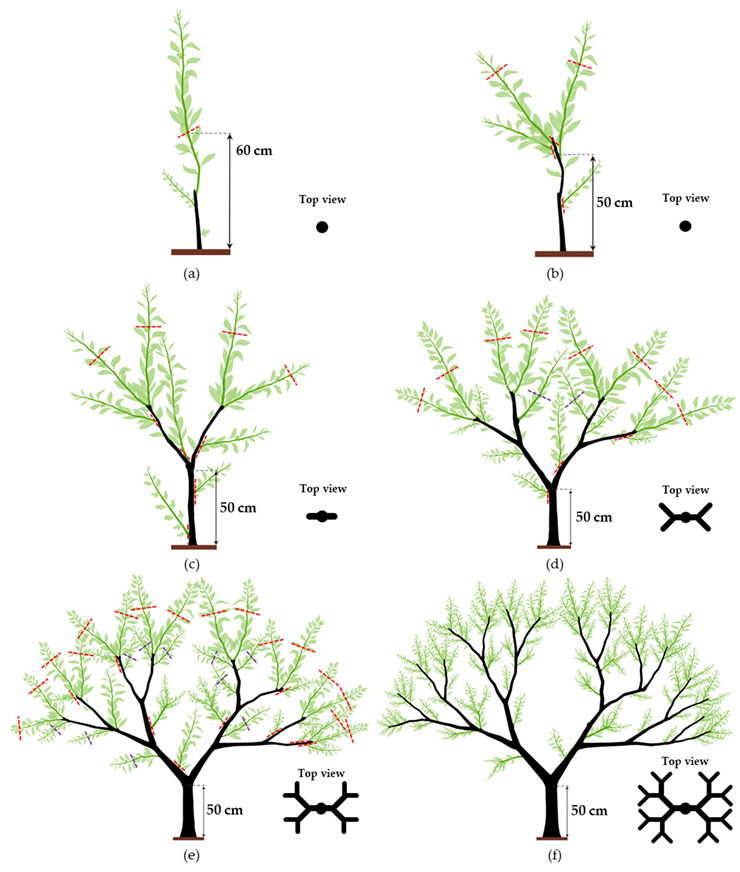
Dichotomic training system scheme of pruning (structure-forming cuts in red) and top view of the structural branches. (**a**) The main stem is cut at the nursery pruning or at planting; (**b**) two branches are chosen to form the first dichotomic level and cut at 50 cm; (**c**) four branches are selected from the first dichotomic level to start the second dichotomic level and cut at 50 cm; (**d**) the branches that will form the third dichotomic level are selected, and some branches can be pruned by using a heading cut (purple lines) to form temporary production branches; (**e**) the fourth dichotomic level is selected; (**f**) the final tree structure presents at least four dichotomic levels.

**Figure 15 plants-12-03360-f015:**
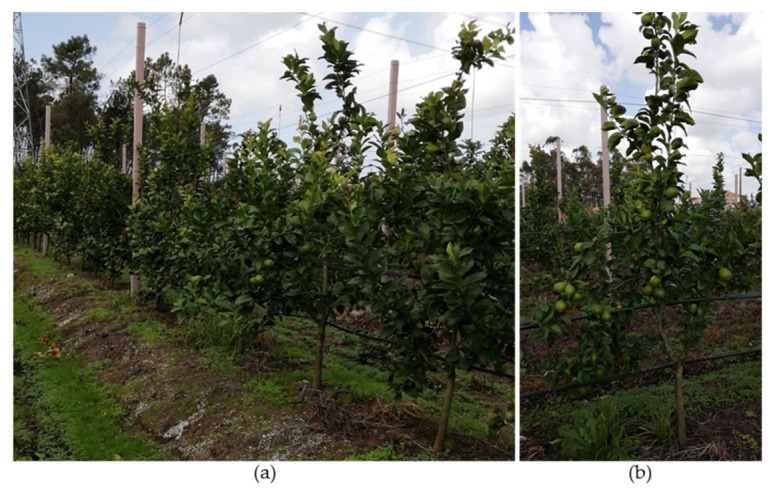
Central leader training system, applied to lemon trees grafted on *Citrus trifoliata*. (**a**) This system is implemented in orchards with high planting densities. (**b**) Side branches grow from the axis. (Photographs by Hugo Marques.).

**Figure 16 plants-12-03360-f016:**
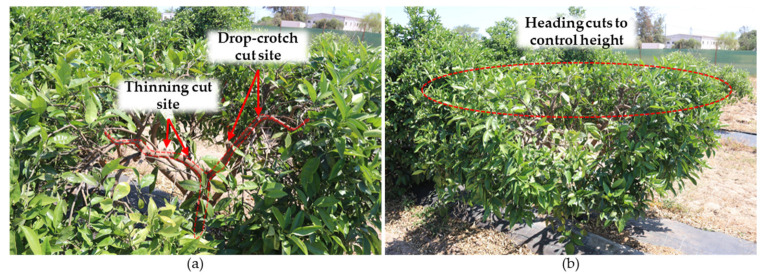
‘Washington Navel’ orange trees trained in an open-center shape. (**a**) Drop-crotch cuts boost side-branch growth and canopy opening; the red arrows show the pruning points of some structural branches. (**b**) Structural branches are maintained at the same height; the red circle highlights the result of the heading cuts done to keep the structural branches at the same height.

**Figure 17 plants-12-03360-f017:**
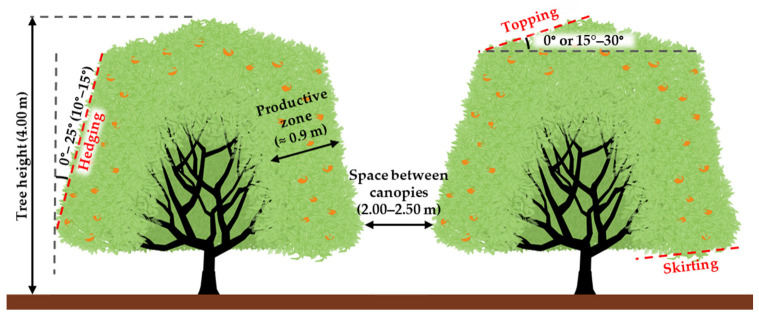
Scheme of the mechanical pruning cutting operations that include hedging, topping, and skirting.

**Figure 18 plants-12-03360-f018:**
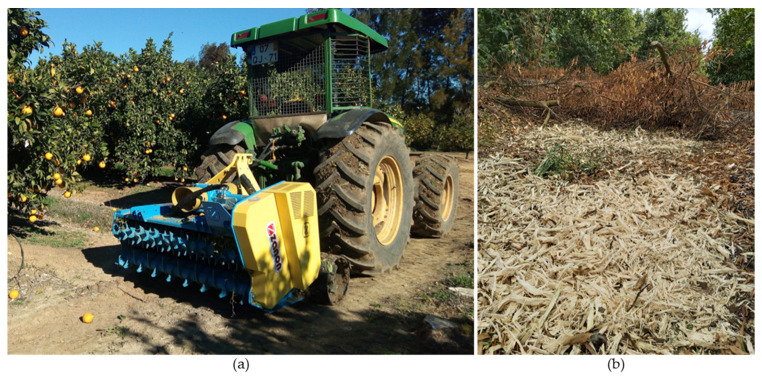
Shredding of pruning residues. (**a**) Shredder working in a citrus orchard. (**b**) Pruned branches before and after shredding.

**Table 1 plants-12-03360-t001:** Brief classification and characterization of citrus cultivars.

Species or Species Group	Group	Cultivar	Tree Growth Habit ^1^	Shooting Habit ^2^	Ripening ^3^	Bearing Habit ^4^
Sweet orange *Citrus sinensis*	Common oranges	‘Valencia Late’	Spreading *	IS	Late	SB/AB
		‘Dom João’	Spreading	IS	Late	SB/AB
		‘Salustiana’	Upright *	IS	Mid-season	SB
	Navel oranges	‘Washington Navel’ *	Spreading	IS	Mid-season *	SB
		‘Navelina’	Spreading *	IS	Early *	SB
		‘Newhall’	Spreading *	IS	Early *	SB
		‘Navelate’	Spreading *	IS	Mid-season *	SB
		‘Powell’	Spreading	IS	Mid-season	SB
		‘Lane Late’	Spreading *	IS	Mid-season *	SB
		‘Rhode Navel’	Spreading	IS	Mid-season	SB
		‘Barnfield’	Spreading *	IS	Mid-season *	SB
	Pigmented or blood oranges	‘Tarocco’	Upright/Spreading	IS	From early to Late	SB
		‘Moro’	Spreading	IS	Mid-season	SB
		‘Sanguinelli’	Spreading *	IS	Mid-season *	SB
Mandarin	Common mandarins *Citrus reticulata* Blanco	‘Setubalense’	Spreading	SMS	Mid-season	AB
		‘Avana’	Spreading	SMS	Early	SB
	Clementines *Citrus clementina* Hort. ex Tanaka	‘Clemenules’	Spreading *	SMS	Mid-season *	SB
		‘Marisol’	Upright *	SMS	Early *	SB
		‘Oronules’	Spreading *	SMS	Early *	SB
		‘Loretina’	Upright *	SMS	Early *	SB
		‘Beatriz’	Spreading *	SMS	Early *	SB
		‘Fina’	Spreading *	SMS	Mid-season *	SB
	Satsumas *Citrus unshiu* (Mak) Marc.	‘Hashimoto’	Drooping *	LSS	Early *	SB
		‘Okitsu’	Drooping *	LSS	Early *	SB
		‘Clausellina’	Drooping *	LSS	Early *	SB
		‘Owari’	Drooping *	LSS	Early *	SB
	Mandarin hybrids and others	‘Afourer’’	Upright	IS	Mid-season	SB/AB
		‘Encore’	Upright	IS/SMS	Mid-season	AB
		‘Nova’	Spreading *	SMS	Mid-season	SB
		‘Ortanique’	Spreading *	IS	Mid-season	SB
Lemon	Lemon *Citrus limon* Burn.	‘Eureka’	Spreading *	LSS	Early *	MB
		‘Verna’	Spreading	LSS	Mid-season	MB
		‘Fino’	Upright	LSS	Early *	MB

**^1^ Tree growth habit:** upright (vertical branches, acute insertion angles, tall canopy), spreading (fewer vertical branches, wider insertion angles, lower canopy), and drooping (horizontal branches, wide insertion angles, low canopies). **^2^ Shooting habit:** LSS (long solitary shoots), IS (intermediate shooting), and SMS (short multiple shoots). **^3^ Ripening:** early (fruit development 7 months or less), mid-season (fruit development between 8 and 12 months) and late (fruit development longer than 12 months). **^4^ Bearing habit** (in subtropical climates): SB (single-annual-bearing), MB (multiple-annual-bearing), and AB (alternate-bearing). * Source: [36].

**Table 2 plants-12-03360-t002:** Pruning intensity according to the purpose *.

Pruning Intensity	Purpose
Very strong; 50% canopy removal	Canopy renovation.
	Preparation of the trees to be eliminated in intensive orchards (temporary trees).
Strong; 30% canopy removal	Improving the inside-canopy illumination. Partial canopy renovation.
	Production regularization in alternate-bearing cultivars in years of abundant flowering.
Normal; 20% canopy removal	Vegetation renewal to maintain tree balance over time.
Light; 10% canopy removal	Production regularization in alternate-bearing cultivars in years of low flowering.
	Annual maintenance pruning in vigorous trees.

* Adapted from Rodríguez and Villalba [5].

**Table 3 plants-12-03360-t003:** Summary table of the different types of pruning in citrus.

Pruning Type	Tree’s Life Stage	Pruning Intensity	Main Purposes
**Formative pruning**	Young tree—at planting and in the first 3 to 5 years of life.	<25%	Forming a branch structure that supports the entire tree canopy. It must be performed according to the chosen training system.
**Maintenance pruning**	Mature tree—when the tree is at full production (approximately from the fifth year after planting).	10–30%	To renew the production branches. To favor good vegetation and fruit distribution. Improving the light distribution and air circulation inside the canopy. Production regulation. Improving fruit size and quality. To control the canopy size.
**Recovery pruning**	Mature trees—if trees present dense and oversized canopies.	30–50%	To reduce canopy height. Improving the light distribution and air circulation inside the canopy. To favor the partial renewal of the canopy.
**Rejuvenation pruning**	Old trees—when production starts to decline and the entire canopy must be renewed.	>50%	To rejuvenate the tree. To change the cultivar, keeping the same structure.

**Table 4 plants-12-03360-t004:** Pruning strategy in alternate-bearing cultivars with different ripening periods.

Fruit Ripening Period of the Cultivar	On-Year (Abundant Flowering)	Off-Year (Scarce Flowering)
Early	30% intensity Early summer, after fruit set (2nd period).	10% intensity After harvest, before flowering (1st period).
Mid-season	30% intensity Early summer, after fruit set (2nd period).	10% intensity (only if necessary) Early summer, after fruit set (2nd period).
Late	30% intensity From fruit drop to August or later (3rd period).	10% intensity (only if necessary) From fruit drop to August or later (3rd period).

**Table 5 plants-12-03360-t005:** Summary table of advantages and disadvantages of manual and mechanical pruning.

Advantages	
Mechanical pruning	Manual pruning
Less expensive. Faster. Less labor-intensive.	Well adapted to the natural growth habit of citrus trees. Hand tools such as pruning scissors and chainsaws are easy to access.
**Disadvantages**	
Mechanical pruning	Manual pruning
Not well adapted to the natural growth habit of citrus. Requires heavy machinery that is sometimes difficult to access. In mid-season and late cultivars, it can lead to yield loss since the cuts are made across the entire productive zone of the canopy. Regular mechanical pruning leads to the formation of dense “foliage walls” with the following disadvantages: Worse canopy aeration. Solar radiation penetration into the canopy decreases. The process reduces the effectiveness of phytosanitary treatments. The process makes it difficult to enter the tree, particularly during harvesting.	More expensive. More time-consuming. Requires a lot of skilled labor. Labor-intensive.

**Table 6 plants-12-03360-t006:** Examples of possible pruning programs.

Strategy	Year 1	Year 2	Year 3	Year 4
**1. THH/THH**	Topping + bilateral hedging	Topping + bilateral hedging	Topping + bilateral hedging	Topping + bilateral hedging
**2. THH/--**	Topping + bilateral hedging	--	Topping + bilateral hedging	--
**3. TH/TH**	Topping + Side 1 hedging	Topping + Side 2 hedging	Topping + Side 1 hedging	Topping + Side 2 hedging
**4. HH/T**	Bilateral hedging	Topping	Bilateral hedging	Topping
**5. H/H/T**	Side 1 hedging	Side 2 hedging	Topping	Side 2 hedging
**6. HH/HH**	Bilateral hedging	Bilateral hedging	Bilateral hedging	Bilateral hedging
**7. T/T**	Topping	Topping	Topping	Topping

## Data Availability

Not applicable.

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
