# Peer review of "Citrus Pruning in the Mediterranean Climate: A Review"

_plants, 2023, doi:10.3390/plants12193360_

Round 1

Reviewer 1 Report

This paper mainly elaborated the main purpose of citrus pruning, summarized and explains the pruning techniques in Mediterranean citrus cultivation, and puts forward the purpose of each pruning type according to the morphology and physiology of citrus. However, there are some problems in this paper.

1. There are too many keywords in the article, and "sweet orange" is mentioned in the keywords, but the article mainly talks about the pruning of citrus, and hardly mentions "sweet orange";

2. The second part "General aspects of citrus morphology and physiology" is too complicated, so it is suggested to delete some contents;

3.The last part of the article lacks a summary part;

4. The format of references in this article is confusing, please check it carefully;

5. The fourth part "Pruning objectives in citrus" should be put behind the pruning principle;

6. There are many formatting problems in the article, such as lines 390, 674 and 701.

Moderate editing of English language required

Author Response

We thank reviewer 1 comments and suggestions that helped us to improve the manuscript quality. Accordingly, we made the corrections as explained below. Reviewer 1 comments are in bold, and our responses are in blue.

  1. There are too many keywords in the article, and "sweet orange" is mentioned in the keywords, but the article mainly talks about the pruning of citrus, and hardly mentions "sweet orange";

We agreed and eliminated "sweet orange" from the keywords.

  1. The second part "General aspects of citrus morphology and physiology" is too complicated, so it is suggested to delete some contents;

We agreed and consequently reduced the chapter content significantly.

3.The last part of the article lacks a summary part;

A final remarks chapter was added.

  1. The format of references in this article is confusing, please check it carefully;

All the references were carefully corrected.

  1. The fourth part "Pruning objectives in citrus" should be put behind the pruning principle;

Following the reviewer's suggestion, we changed the "Pruning objectives in citrus" and "Pruning fundamentals" chapter order

  1. There are many formatting problems in the article, such as lines 390, 674 and 701.

We carefully reviewed all the paper and corrected the formatting problems.

Reviewer 2 Report

Manuscript Number: plants-2492877

Title: Citrus pruning in the Mediterranean climate: a review

The review entitledCitrus pruning in the Mediterranean climate: a review» will address general aspects of citrus pruning. It will focus in more detail on the practices used in Mediterranean countries, according to the authors' experience after several years of work on pruning trials. The manuscript is not well writing, the objectives are not clear. The discussion is not appropriated.

English very difficult to understand/incomprehensible

English very difficult to understand/incomprehensible

Author Response

We thank reviewer 2 for the comments on our paper entitled “Citrus pruning in the Mediterranean climate: a review”. Accordingly, we improved the writing and made the objectives clearer. The discussion was also improved.

Round 2

Reviewer 1 Report

Accept

Fine

Reviewer 2 Report

no comments